# Enhancing Contrastive Learning for Retinal Imaging via Adjusted Augmentation Scales

**Cheng, Zijie**[1]                                    ZIJIE.CHENG.23@UCL.AC.UK
[1] *UCL Department of Medical Physics & Biomedical Engineering, United Kingdom*

**Li, Boxuan**[1]                                      B.LI.22@UCL.AC.UK
**Altmann, Andre**[1,2]                                A.ALTMANN@UCL.AC.UK
[2] *UCL Hawkes Institute, United Kingdom*

**Keane, Pearse**[3]                                   P.KEANE@UCL.AC.UK
[2] *UCL Institute of Ophthalmology, United Kingdom*

**Zhou, Yukun**[2,3]                                   YUKUN.ZHOU.19@UCL.AC.UK

**Editors:** Accepted for publication at MIDL 2025

## Abstract

Contrastive learning, a typical self-supervised learning strategy, operates on bringing similar data together while pushing dissimilar data apart in latent space. This approach extracts robust and discriminative representations, thus being widely used in natural computer vision tasks, such as object classification. However, unlike natural images, medical images (e.g., retinal images) tend to share substantial similarities in imaging area and anatomical tissues, leading to a denser distribution in latent space. As a result, the default use of strong augmentations in contrastive learning potentially exacerbates this intensive distribution in retinal images, making it difficult to distinguish between genuinely similar and dissimilar data, and therefore hindering model pre-training convergence. In this paper, we hypothesise that weaker augmentations are better suited to contrastive learning for medical image applications, and we investigate model performance under various augmentation strategies. Our study includes six publicly available retinal datasets covering multiple clinically relevant tasks. We assess the models' performance and generalizability via extensive experiments. The model pre-trained with weak augmentation outperforms the one pre-trained with strong augmentation, achieving approximately a 6% increase in AUPR ($P<0.001$) and a 12.5% increase in sensitivity ($P<0.001$) on MESSIDOR-2. Similar improvements are observed across other datasets. Our findings suggest that optimizing the scale of augmentation is critical for enhancing the efficacy of contrastive learning in medical imaging. The model weights and relevant code are available at: https://github.com/ziijiecheng/Enhance-contrastive-SSL-for-Retinal-Imaging.

**Keywords:** contrastive learning, augmentation scales, data distribution, retinal imaging

## 1. Introduction

Contrastive learning is a machine learning paradigm that pulls similar data points (e.g., images rotated from the same image) closer and pushes dissimilar ones (e.g., images rotated from different images) farther apart in the latent space without relying on explicit labels (Lê Khác et al., 2020; Jaiswal et al., 2020). Such an approach trains the model to learn generalizable features. Although pre-trained only on unlabeled data, the models

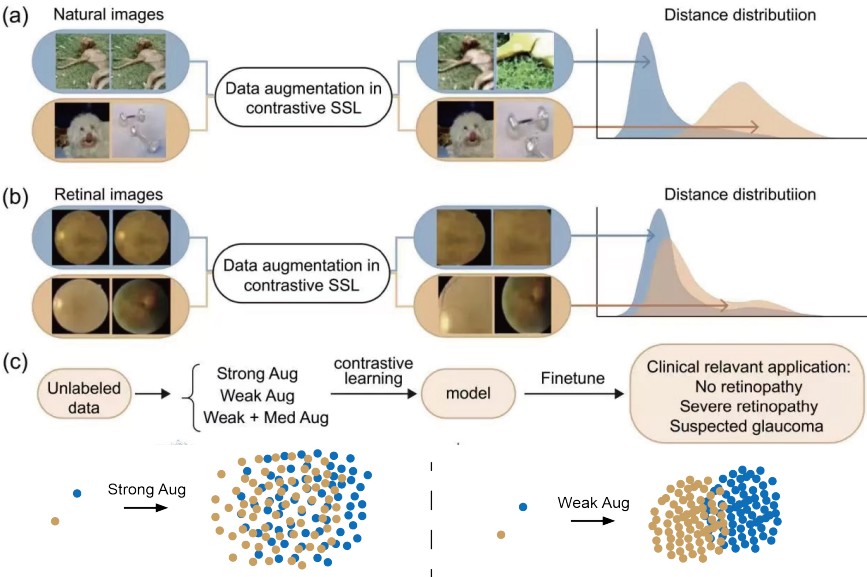

Figure 1: Figures (a) and (b) illustrate the distribution of distances between positive pairs and negative pairs in both natural and medical image domains. Figure (c) presents the project pipeline: unlabeled data is used to pre-train contrastive learning models while investigating various augmentation strategies. The blue dots and yellow dots indicate augmented images from different original images. The goal of this approach is to enhance feature clustering and improve the accuracy of retinal disease classification.

have demonstrated comparable or even better performance compared to supervised learning-based methods (Misra and Maaten, 2019; Hendrycks et al., 2019). In natural image domain, contrastive learning has achieved promising results in diverse tasks such as object detection (Xie et al., 2021), image classification (Zeng and Xie, 2020), and video analysis (Singh et al., 2021). Compared to generative learning, contrastive learning has shown better effectiveness in various applications (Oquab et al., 2024; Caron et al., 2021; Liu et al., 2020). However, whether this observation extends to medical images remains underexplored.

Recent research has started comparing contrastive learning and generative learning in medical artificial intelligence (AI). For instance, RETFound (Zhou et al., 2023), a foundation model for retinal images, employed a generative learning strategy named the Masked Autoencoder (He et al., 2021) for model development and demonstrated superior performance compared to contrastive learning methods in retinal disease classification. Understanding the reasons behind this inconsistency and developing a simple yet efficient solution to improve contrastive learning for medical imaging is crucial.

The suboptimal performance of contrastive learning in medical imaging is likely due to inherent differences between the distributions of natural and medical images (Wen et al., 2021). Natural images are colorful with varying pixel intensities, while medical images are usually grayscale and structurally similar, especially within the same organ or tissue type (Legras et al., 2018; Durston et al., 2001). This characteristic results in a denser distribution of medical images within the latent space compared to natural images (Zhou et al., 2021). We

hypothesize that such a dense distribution degrades performance when applying contrastive learning methods to medical images. As shown in Figures 1(a) and 1(b), natural images under the default strong augmentations in contrastive learning are sparsely distributed in latent space, while different medical images tend to overlap heavily. The pretext task of contrastive learning aims to distinguish between positive pairs (augmented views of the same image) and negative pairs (augmented views of different images). In the context of medical imaging, the significant overlap of augmented images in the latent space renders this pretext task highly challenging, thereby hindering model converge in contrastive learning. Due to the unique characteristics of medical images, previous studies have explored specific augmentation methods for different medical modalities (Goceri, 2023; van der Sluijs et al., 2023; Kang et al., 2022).

In this work, we propose a simple yet effective solution to enhance contrastive learning performance by reducing augmentation scales. The project pipeline is illustrated in Figure 1(c). We use Distillation with No Labels (DINO) (Caron et al., 2021) as a study example of contrastive learning strategies, and validated our solution on clinical applications, including glaucoma and diabetic retinopathy diagnosis, using both internal and external evaluations. Our approach not only enhances feature clustering but also demonstrates improved diagnostic performance compared to models using default strong augmentations.

## 2. Methods

### 2.1. Problem Definition

For contrastive learning, given a set of unlabeled retinal images $\mathcal{D} = \{x_i\}_{i=1}^{N}$, we create positive pairs $\mathcal{P}^+$ by randomly selecting an image $x_i \in \mathcal{D}$ and apply twice augmentation $\Phi_{t,s}$ respectively to get augmented data $x_i^1$ and $x_i^2$, where $t$ indicates the augmentation type and $s$ the scale range. While for negative pairs, we sample two images $x_i$ and $x_j \in \mathcal{D}$ (with $i \neq j$) and apply the augmentation to each image, forming the negative pair $\mathcal{P}^- = (x_i^1, x_j^2)$. We then use the feature encoder from model $f$ to project these images in latent space, such as $En(x_i^1)$. The distance between positive pairs and negative pairs in latent space can be measured by $Dis(\cdot)$:

$$\text{Dis}(\mathcal{P}^+) = \left\| En(x_i^1) - En(x_i^2) \right\|_2 = \sqrt{\sum_{k=1}^{d} \Big( En(x_i^1)_k \ - \ En(x_i^2)_k \Big)^2}, \tag{1}$$

$$\text{Dis}(\mathcal{P}^-) = \left\| En(x_i^1) - En(x_j^2) \right\|_2 = \sqrt{\sum_{k=1}^{d} \Big( En(x_i^1)_k \ - \ En(x_j^2)_k \Big)^2}, \tag{2}$$

where $En(\cdot)$ maps an image into a latent space (i.e., an embedding) with $d$ as the dimension of the latent representation. The index $k$ ranges from 1 to $d$, with $En(x_i^1)_k$ denoting the $k$-th component of the embedding vector $En(x_i^1)$.

The general training objective of contrastive learning is to train the model $f$ to maximize the distance between negative pairs and to minimize that for positive pairs,

$$f = \text{argmax}\left( \text{Dis}(\mathcal{P}^-) - \text{Dis}(\mathcal{P}^+) \right). \tag{3}$$

When $\text{Dis}(\mathcal{P}^+)$ approximates $\text{Dis}(\mathcal{P}^-)$, it is challenging to train the model $f$ to converge well. This issue is prominent in medical imaging due to less variation compared to natural images. For example, retinal images depict the anatomical tissue of retina, often showing similar structure and orientation (Patton et al., 2006), as shown in Figure 1(b). With strong augmentations $\Phi_{strong}$ (e.g., cropping the images into small patches) following the default augmentation settings in DINO, $\text{Dis}(\mathcal{P}^-)$ decrease while $\text{Dis}(\mathcal{P}^+)$ increases, which brings further challenges in achieving objects of equation 3 and may result in suboptimal model pre-training with contrastive learning, showing the poor performance in classifying the positive and negative pairs.

Such suboptimal model performance extends to downstream applications, where models are fine-tuned with labeled data $\mathcal{D}_l = \{x_i, y_i\}_{i=1}^{L}$ for diverse tasks like disease diagnosis, where $x$ represents the data and $y$ indicates the label. To improve the model's capability in clinically meaningful applications, our strategy involves enhancing the contrastive learning performance in classifying $\mathcal{P}^+$ and $\mathcal{P}^-$ by specifically decreasing $\text{Dis}(\mathcal{P}^+)$ while increasing $\text{Dis}(\mathcal{P}^-)$.

### 2.2. Scattering Data Distribution with Weak Augmentations

To achieve such a goal for retinal images, a straightforward strategy is to scale down the augmentation. An extreme case is to remove the augmentation so that $\text{Dis}(\mathcal{P}^+)$ achieves 0 and $\text{Dis}(\mathcal{P}^-)$ stays as a high value. However, pre-training without any augmentation hardly trains the model to learn generalizable and diverse features. Hence, we propose to scale down the augmentation, termed $\Phi_{weak}$, to ease the challenge of training the model $f$ to converge while also avoiding it being too weak for the model to learn generalizable features. Additionally, we investigate the effects of several augmentations that mimic the retinal image artefacts, including random bias field and Gaussian blur. We combine it with $\Phi_{weak}$ to form $\Phi_{weak+med}$.

## 3. EXPERIMENT

### 3.1. Data

The pre-training data are from Moorfields Eye Hospital (Wagner et al., 2022; Zhou et al., 2023) with 1.4 million color fundus images, a retinal image modality. These images were collected from a retrospective cohort study linking ophthalmic data of 353,157 patients, who attended the hospital between 2008 and 2018. All images are preprocessed and resized to $224 \times 224$ by an automated retinal image analysis tool AutoMorph (Zhou et al., 2022).

We evaluate the efficacy of different augmentation strategies using clinically meaningful tasks, including diabetic retinopathy (DR) diagnosis, glaucoma detection, and multi-class retinal disease classification. For DR diagnosis, we include MESSIDOR-2 (Decencière et al., 2014), IDRiD (Porwal et al., 2018), and APTOS2019 (APTOS, 2019). The labels for DR are based on the International Clinical DR Severity Scale, covering five stages from no DR to proliferative DR. For glaucoma diagnosis, we use the PAPILA dataset (Kovalyk et al., 2022), which has three categorical labels: non-glaucoma, early glaucoma (suspected glaucoma), and advanced glaucoma. For multi-class disease classification tasks, we use two datasets, JSIEC (Cen et al., 2021) containing 1,000 images with 39 categories of common retinal diseases and

Table 1: Data summary for the datasets used for disease diagnosis. Each dataset is split into training, validation, and testing sets.

| Dataset | Country | Categories | Training | Validation | Testing |
|---------|---------|------------|----------|------------|---------|
| **Diabetic retinopathy** | | | | | |
| MESSIDOR-2 | France | 5 | 972 | 246 | 526 |
| IDRiD | India | 5 | 329 | 84 | 103 |
| APTOS2019 | India | 5 | 2048 | 514 | 1100 |
| **Glaucoma** | | | | | |
| PAPILA | Spain | 3 | 312 | 79 | 98 |
| **Multi-class disease** | | | | | |
| JSIEC | China | 39 | 534 | 150 | 316 |
| Retina | NR | 4 | 336 | 84 | 181 |

conditions, and Retina dataset (jr2ngb, 2023) with labels for normal, glaucoma, cataract, and retinal disease. Data splitting details are shown in Table 1.

### 3.2. Pre-training details

DINO (Caron et al., 2021), a representative and commonly used contrastive learning strategy, was used in the experiment. We first initialized the model with ImageNet weights and then pre-trained it using 1.4 million color fundus images from Moorfields Eye Hospital. The data preprocessing, data quality control, model architecture, and hyperparameters (except for those related to augmentations) were standardized to ensure a fair comparison. The model was pre-trained using an NVIDIA A100 (80G). The details of $\Phi_{strong}$, $\Phi_{weak}$, and $\Phi_{weak+med}$ are listed in Table 2. $\Phi_{strong}$ follows the default augmentation settings in DINO, which was well-tuned on natural images. Local crop is a small, zoomed-in region of an image. Global Crop is a large region of an image. Color jittering involves random adjustments to image brightness and contrast. Gaussian blur is a smoothing effect created by applying a Gaussian filter to reduce detail and noise. Gaussian noise consists of random intensity variations that follow a Gaussian distribution. Random bias field is a smooth, spatially varying intensity variation across an image. The $\Phi_{med}$ introduced into $\Phi_{weak}$ is implemented from Torchio, a Python library for medical image processing (Pérez-García et al., 2021).

We then compared these models by adapting them to downstream tasks of disease diagnosis. We evaluated the model performance with the Area Under the Receiver Operating Characteristic curve (AUROC), the Area Under the Precision-Recall curve (AUPR), and sensitivity. Each experiment is run five times with random seeds to obtain performance statistics.

### 3.3. Experiment Result

We first plotted the distribution of distances between positive pairs and distances between negative pairs in different augmentation strategies. The model pre-trained with $\Phi_{weak}$ better distinguished these pairs, as shown in Figure 2(a). We also observed the clustering performance of the models, that is, how positive and negative pairs were distributed, across different augmentation strategies through the t-SNE map (Maaten and Hinton, 2008). We

Table 2: Various settings of augmentation types and scales. Augmentations not listed are consistent with the default strong augmentations, well-tuned on natural images. For local and global crops, the range (e.g., (0.05, 0.4)) represents the cropping scales relative to the original image. The symbol $p$ denotes the probability of applying a particular transformation, which is defaulted as 1 unless specified. $\times$ indicates that the transformation is not applied.

|  | Local crop | Global crop | Color jitter | Blur | Noise | Bias field |
|---|---|---|---|---|---|---|
| $\Phi_{\mathrm{strong}}$ | (0.05, 0.4) | (0.4, 1.0) | bright:0.4 contrast:0.4 | $\times$ | $\times$ | $\times$ |
| $\Phi_{\mathrm{weak}}$ | (0.2, 0.5) | (0.5, 1.0) | bright:0.2 contrast:0.2 | $\times$ | $\times$ | $\times$ |
| $\Phi_{\mathrm{weak+med}}$ | (0.2, 0.5) | (0.5, 1.0) | bright:0.2 contrast:0.2 | std:0.1 $p$:0.5 | std:0.1 $p$:0.5 | scale:0.1 $p$:0.5 |

repeatedly augmented each image to create image groups, where positive pairs consisted of images within the same group, and negative pairs were images from different groups. Then, we projected these images into latent space and found that the features of negative pairs have a distinct distribution under the weak augmentation shown in Figure 2(b). We also used the Silhouette score (Shahapure and Nicholas, 2020) to quantify the clustering quality of DINO pre-trained under different augmentation strategies. DINO pre-trained with $\Phi_{weak}$ achieved the highest score of 0.201, while those pre-trained with $\Phi_{strong}$ and $\Phi_{weak+med}$ achieved scores of 0.117 and 0.130, respectively.

In the internal evaluation presented in Table 3, DINO with $\Phi_{weak}$ outperformed the other augmentation strategies on most retinal disease classification tasks. Specifically, on MESSIDOR-2, PAPILA, JSIEC, and Retina, the model employing $\Phi_{weak}$ consistently demonstrated higher AUROC, AUPR, and sensitivity compared with $\Phi_{strong}$. Notably, on JSIEC, the model pre-trained with $\Phi_{weak}$ achieved a 10% increase in AUPR and a 23.7% increase in sensitivity compared to $\Phi_{strong}$ ($P$<0.001). However, on IDRiD, although the model achieved higher AUROC and AUPR under $\Phi_{weak}$ than under $\Phi_{strong}$, $\Phi_{strong}$ conferred a slight advantage of approximately 1.1% in sensitivity. Introducing the medical augmentation $\Phi_{med}$ generally diminished the model's performance. For example, on IDRiD and JSIEC, combining $\Phi_{weak}$ with $\Phi_{med}$ ($\Phi_{weak+med}$) reduced performance, particularly sensitivity, by 6.4% and 12.5% compared to $\Phi_{weak}$, respectively. These tasks often show low sensitivity, as seen in Diabetic Retinopathy classification with five classes, a common challenge in this application (Islam et al., 2022; Long et al., 2024).

As shown in Table 4, the external evaluation indicated that $\Phi_{weak}$ performed better than $\Phi_{strong}$ and $\Phi_{weak+med}$ in most tasks. For instance, when the model fine-tuned on IDRiD was externally evaluated on APTOS2019 and MESSIDOR-2, the model pre-trained with $\Phi_{weak}$ outperformed $\Phi_{strong}$ by 0.4% and 1.7%, respectively.

## 4. Discussion and Conclusion

In this study, we aimed to improve the contrastive learning performance in the medical image domain. We proposed a hypothesis that the dense distribution of medical images might cause the suboptimal performance of contrastive learning, and validated it in our experiments.

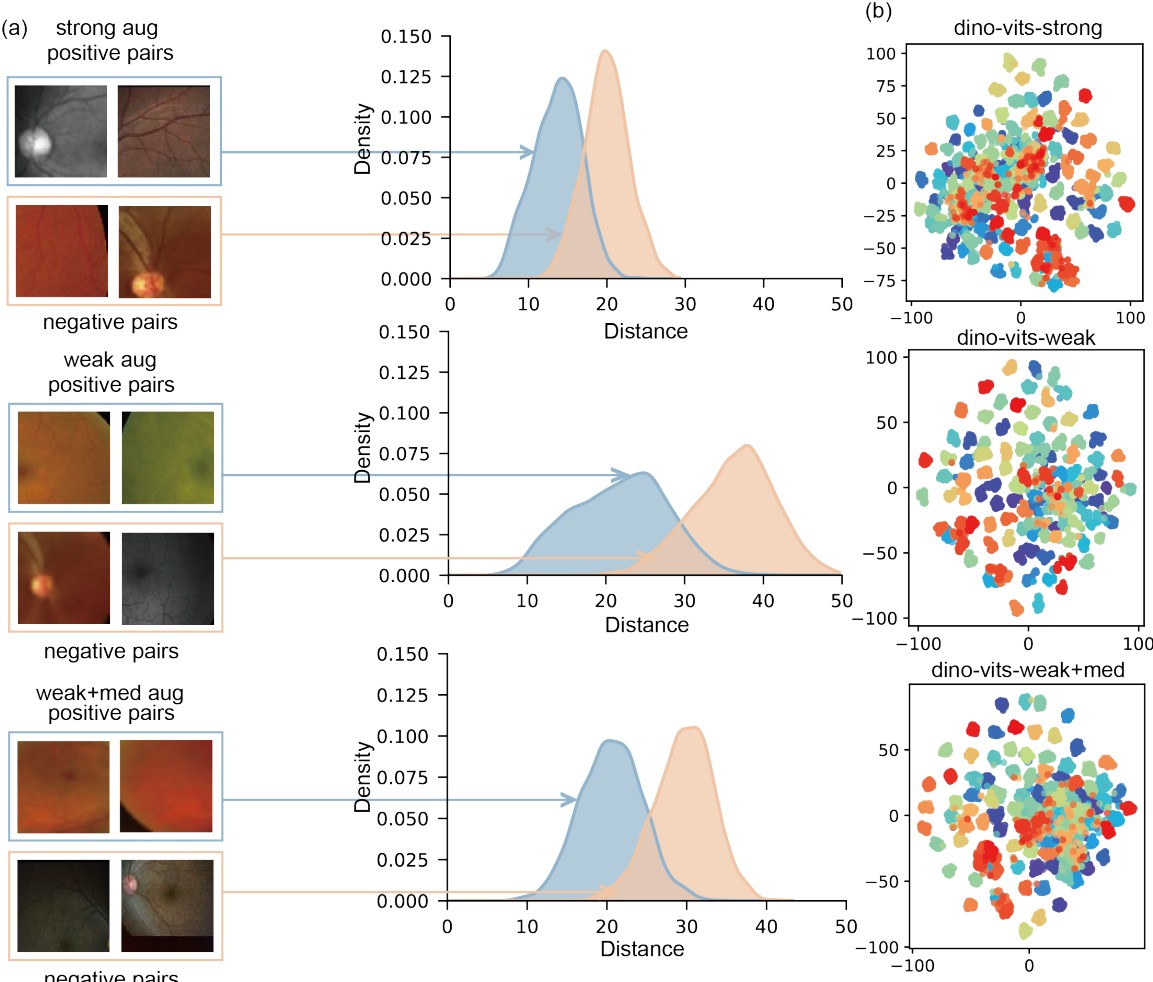

Figure 2: We extract features using the DINO teacher model (encoder), pre-trained respectively with $\Phi_{strong}$, $\Phi_{weak}$ and $\Phi_{weak+med}$. First, we calculate the Euclidean distances between positive and negative pairs and compare their distance distributions in Figure (a). We also use a t-SNE map to visualize feature clustering in latent space in Figure (b), where different colors represent augmented views from different images. In figure (b), vits represents small ViT.

Our findings suggest that simply reducing augmentation scales to an appropriate level can improve the clustering performance and therefore enhance model performance in downstream tasks. Additionally, when incorporating medical-specific augmentation $\Phi_{med}$ to $\Phi_{weak}$, the collective augmentation could again decrease $\text{Dis}(\mathcal{P}^-)$, while increase $\text{Dis}(\mathcal{P}^+)$ (Figure 2), generating adverse effects on model performance. These offer key guidance into the model pre-training with contrastive learning for medical images.

Although bringing insights, we acknowledge several limitations in this work that should be studied in future work. First, the performance under $\Phi_{weak}$ sometimes only has a slight

Table 3: Model comparison on disease diagnosis with internal evaluation. The middle three columns show model performance under varied data augmentation strategies, with the highest value in each row highlighted in bold. For each task, the model was fine-tuned using five random seeds (affecting training data shuffling) and evaluated on the test set, yielding five replicas. Statistical significance was determined via a repeated-measures analysis of variance (ANOVA), with random seed treated as a within-subjects factor. The resulting $P$ values quantify the significance of performance differences between $\Phi_{weak}$ and $\Phi_{strong}$.

| | $\Phi_{strong}$ | $\Phi_{weak}$ | $\Phi_{weak+med}$ | $P$ value |
|---|---|---|---|---|
| **MESSIDOR-2** | | | | |
| AUROC | .838 (.835, .840) | **.848 (.846, .851)** | .823 (.817, .829) | $< .001$ |
| AUPR | .523 (.516, .530) | **.582 (.575, .589)** | .523 (.498, .547) | $< .001$ |
| Sensitivity | .154 (.146, .162) | **.279 (.255, .303)** | .247 (.220, .274) | $< .001$ |
| **APTOS2019** | | | | |
| AUROC | .933 (.932, .933) | **.933 (.933, .934)** | .924 (.924, .925) | .004 |
| AUPR | **.667 (.665, .670)** | .665 (.661, .668) | .637 (.635, .639) | .236 |
| Sensitivity | .469 (.466, .472) | **.528 (.509, .547)** | .482 (.474, .491) | .003 |
| **IDRiD** | | | | |
| AUROC | .747 (.736, .758) | **.790 (.782, .798)** | .726 (.720, .732) | $< .001$ |
| AUPR | .461 (.445, .476) | **.498 (.486, .509)** | .432 (.419, .446) | $< .001$ |
| Sensitivity | **.366 (.343, .390)** | .355 (.340, .369) | .291 (.266, .316) | .369 |
| **PAPILA** | | | | |
| AUROC | .791 (.782, .800) | **.816 (.804, .829)** | .792 (.785, .800) | .003 |
| AUPR | .637 (.630, .643) | **.671 (.653, .688)** | .628 (.619, .638) | .018 |
| Sensitivity | .238 (.205, .271) | .295 (.264, .327) | **.312 (.282, .341)** | .096 |
| **JSIEC** | | | | |
| AUROC | .960 (.958, .962) | **.977 (.975, .979)** | .968 (.967, .969) | $< .001$ |
| AUPR | .651 (.637, .664) | **.760 (.750, .769)** | .707 (.695, .720) | $< .001$ |
| Sensitivity | .331 (.316, .345) | **.568 (.557, .578)** | .443 (.416, .470) | $< .001$ |
| **Retina** | | | | |
| AUROC | .781 (.776, .787) | .807 (.801, .813) | **.814 (.807, .820)** | $< .001$ |
| AUPR | .594 (.585, .604) | **.632 (.620, .643)** | .626 (.616, .635) | .002 |
| Sensitivity | .326 (.321, .332) | **.416 (.396, .436)** | .375 (.352, .399) | $< .001$ |

advantage compared to that under $\Phi_{strong}$ in both internal and external evaluation. This is likely caused by nearly saturated performance after pretraining on large-scale nature images. Some techniques, such as methods for automatically adjusting augmentation scales will be studied to achieve optimised performance. Specifically, for positive pairs that are too far apart in latent space and negative pairs that are too close, the loss function will assign greater weights to them during model pre-training. Second, we only validated our hypothesis and solution on DINO; more contrastive learning strategies, such as DINOv2 (Oquab et al., 2024), could be investigated. Third, some quantitative metrics describing the clustering performance have not been investigated, which will be proposed in future work to guide the augmentation scaling. This work pioneered the optimization of contrastive learning in the medical domain and encouraged tailored model training settings for medical images.

Table 4: This table presents the external evaluation results on diabetic retinopathy datasets based on AUROC. For each dataset pair, the highest mean value among the different augmentation strategies is highlighted in bold. For each task in interval evaluation, we generate five replicas using different random seeds. For each replica, the model weights corresponding to the best performance on the validation set are saved for subsequent external performance assessment. Statistical significance was determined via a repeated-measures analysis of variance (ANOVA), with random seed treated as a within-subjects factor. The resulting $P$ values quantify the significance of performance differences between $\Phi_{weak}$ and $\Phi_{strong}$.

| Fine-tune data | APTOS2019 | | IDRiD | | MESSIDOR-2 | |
|---|---|---|---|---|---|---|
| **Test data** | IDRiD | MESSIDOR-2 | APTOS2019 | MESSIDOR-2 | APTOS2019 | IDRiD |
| $\Phi_{strong}$ | $.785 \pm .006$ | $\mathbf{.767 \pm .001}$ | $.740 \pm .011$ | $.744 \pm .008$ | $\mathbf{.804 \pm .007}$ | $.743 \pm .010$ |
| $\Phi_{weak}$ | $\mathbf{.790 \pm .005}$ | $.761 \pm .002$ | $\mathbf{.744 \pm .012}$ | $\mathbf{.761 \pm .007}$ | $.798 \pm .009$ | $\mathbf{.746 \pm .007}$ |
| $\Phi_{weak+med}$ | $.752 \pm .004$ | $.692 \pm .003$ | $.732 \pm .011$ | $.723 \pm .008$ | $.708 \pm .018$ | $.738 \pm .009$ |
| $P$ value | $.125$ | $<.001$ | $.179$ | $<.001$ | $.211$ | $.316$ |

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
