# OpenReview forum: "Enhancing Contrastive Learning for Retinal Imaging via Adjusted Augmentation Scales"
_MIDL.io/2025/Conference — MIDL 2025 Poster_

### Official Review · Reviewer_BvG5 · 2025-02-19

**Confidence:** 4
**Preliminary Rating:** 4
**Recommendation:** Poster
**Final Rating:** 4

**Summary:**

In this paper, they propose a simple yet effective method to enhance contrastive learning for retinal images by adjusting the image augmentation scales. They qualitatively and quantitatively show improved performance for internal and external evaluation on several classification tasks on public datasets compared to the strong augmentations typically applied to natural images.

**Strengths:**

- The paper is well structured, well written, and the figures are well prepared.
- The authors provided a public code repository and evaluated their approach on six publicly available retinal datasets.
- They have extensive experiments and show for some datasets strong performance improvements. For example, for the internal evaluation on MESSIDOR-2, they show a 6% increase in AUPR and a 12.5% increase in sensitivity.
- They trained multiple (5) model variants with different random seeds for each experiment, allowing a statistical significance test based on repeated-measures analysis of variance (ANOVA).

**Weaknesses:**

- As the authors pointed out in the discussion, the weak augmentations sometimes have only a small advantage over the strong augmentations. Moreover, they fixed the scale for the weak augmentations to one variant. Therefore, I'm missing an additional experiment to analyze an optimal augmentation scale, which might be different for different datasets.
- The medical augmentations (random bias field, Gaussian blur, Gaussian noise) didn't seem to help much. Only in two experiments the authors can show improvements with the medical augmentations. There is no analysis, just a discussion of why this might be the case.
- Figure 2b: I'm missing a quantitative clustering metric.
- The results for the weak augmentations are not so strong for the external evaluation compared to the internal evaluation.
- In my opinion, in the introduction the connection to the adaptation of the augmentations themselves and not only their scale to medical imaging is missing, e.g. https://link.springer.com/article/10.1007/s10462-023-10453-z, https://arxiv.org/abs/2301.12636 (X-ray), https://arxiv.org/abs/2212.04690 (histopathology).

**Detailed Comments:**

Figure 2: what does "vits" mean in "dino-vits-"?

**Justification Of The Final Rating:**

Thank you for addressing my concerns and for the extra effort you have put into your submission. My main concerns regarding the fixed scale for the weak augmentations to one variant and the lack of an additional experiment to analyze an optimal augmentation scale, which may be different for different datasets, remain. Therefore, I maintain my preliminary rating of "weak accept".

**Justification Of The Preliminary Rating:**

Using a simple but effective method and extensive experiments, the authors demonstrate the need to adapt contrastive learning methods from machine learning with natural images to medical imaging. In my opinion, this work lacks some analysis to fully understand the effect of weaker augmentation, but it is nevertheless an important step towards an appropriate adaptation of contrastive learning methods to medical images.

**Questions To Address In The Rebuttal:**

- Why do the medical augmentations not help in most cases?
- Why are the results of the external evaluation not so strong?
- Quantitative metric for clustering in Figure 2b.
- Please add references to the related work in the introduction that discuss the adaptation of the augmentations themselves.

**Special Issue:**

No

---

> ### Author Response · Authors · 2025-03-07
>
> Dear reviewer BvG5:
>
> Thank you for your suggestions and for recognizing the efficacy of our method and the well-structured presentation of our manuscript. We carefully read your suggestions and hoped that our explanations addressed your concerns. If any further questions or clarifications were needed, we were happy to provide additional details.
> 1. That is a good question. We discussed this in the first paragraph of the Discussion and Conclusion section. We believe that adding extra medical augmentation to weak augmentation increases the augmentation scale, making the distance between positive pairs larger. However, we will investigate the effect of medical augmentation separately from weak augmentation in the future.
>
> 2. In Table 4, 4/6 results show better performance with weak augmentation, while 2/6 show better performance with strong augmentation. When evaluating the generalizability of a medical AI model using external evaluation, performance usually drops compared to internal evaluation, which reduces the benefit of weak augmentation. Since the advantage of weak augmentation over strong augmentation is not consistent, we will investigate the underlying principles behind this in future work. The weak augmentation scale settings are not the most optimal values. This work aims to validate our hypothesis but does not necessarily require finding the best parameter settings.
>
> 3. That’s a good suggestion! We used the Silhouette score (https://ieeexplore.ieee.org/abstract/document/9260048) to evaluate the clustering of feature embeddings in the latent space. The score under strong augmentation is 0.117, under weak augmentation is 0.201, and under weak+med augmentation is 0.130. We added it to the first paragraph of the Experiment Results section.
>
> 4. Thank you for your suggested references; we have added the ones you recommended to the third paragraph of the Introduction section.
>
> 5. Indeed, we had not clarified it in the manuscript before. It refers to the small vit (as there are small and large vit with different parameter sizes). We have now clarified this in the caption of Figure 2.
>
> We hope you can raise your score, so we can make better contributions to this community. If there are any further suggestions regarding this work, we are very willing to explain and provide more details.

---

> > ### Comment · Reviewer_BvG5 · 2025-03-12
> > **Final Decision**
> >
> > Thank you for addressing my concerns and for the extra effort you have put into your submission. I do not have any additional questions. However, my main concerns regarding the fixed scale for the weak augmentations to one variant and the lack of an additional experiment to analyze an optimal augmentation scale, which may be different for different datasets, remain. Therefore, I am inclined to maintain my preliminary rating of weak acceptance.

---

> > > ### Author Response · Authors · 2025-03-14
> > >
> > > Thank you very much for engaging in the discussion. We are glad to have addressed your concerns. In future extensions, we will explore automatic scaling for data augmentations across multiple medical fields and datasets.

---

### Official Review · Reviewer_QQYm · 2025-02-21

**Confidence:** 5
**Preliminary Rating:** 1
**Final Rating:** 2

**Summary:**

In this work, the author presented some experiments about how the scale of data augmentation affect the performance of the contrastive learning scheme. They made an assumption that: due to the similarity in image color, medical images are not compatible with aggressive augmentation techinique. To prove this hypothesis, they compare the performance of downstream tasks such as DR diagnosis and glaucoma detection. Although the results seem to be supportive, the general idea can be problematic and I don't see any potential to further extend this work to other applications.

* First of all, there is no clear definition of 'strong augmentation' and 'weak augmentation'. The major factor presented in the paper is the cropping ratio. 0.05 is regarded as strong while 0.2 is weak. And other settings for color jitter looks pretty random. More parameters need to be test on to set the boundary between strong and weak.

* More importantly, the whole experiment setting is questionable. During contrastive learning, the positive pairs are those patches cropped from the same image while the negative parts are from different source images. Given that the color jitter is also applied, there is hardly any structural similarity (e.g., retinal vessels) / clinical features (e.g., hemorrhages / lesions) shared between very small patches (0.05) in the same image. I don't see the point of setting them as positive pairs and expect the model to learning useful information during pre-training,

**Strengths:**

This work is trying to improve the implementation of the DINO framework on retinal color fundus images. And this manuscript contains sufficient visualizations which make it easier to follow the idea and interpret the outputs.

**Weaknesses:**

* As it is indicated in the summary, the setting of the augmentation seems to be problematic. Aggressive cropping is incompatible with the retinal color fundus images. During contrastive learning, the positive pairs are those patches cropped from the same image while the negative parts are from different source images. Given that the color jitter is also applied, there is hardly any structural similarity (e.g., retinal vessels) / clinical features (e.g., hemorrhages / lesions) shared between very small patches (0.05) in the same image. Therefore, it is quite natural that the model cannot converge under such condition. I don't see the value of discussing the valid cropping range for fundus images, it is just a ill-posed setting.

* The definitions of 'strong augmentation' and 'weak augmentation' are not rigorous. It is more like the author tried two values, one can work  while the other cannot.

* The evaluation is insufficient. The author only provide the comparison between three different settings of the augmentation during pre-training. How are these results compared with the current state-of-the-art of each downstream tasks?

* Some images created for illustration are not ideal. For example, in Fig.1 (c), it is stated in the caption: 'The blue dots and yellow dots indicate augmented images from different original images'. However, the image seems to show a merging process of two clusters. It should be one blue dot and one yellow dot at the left hand side of the arrow representing two original images from my point of view.

**Detailed Comments:**

* A small suggestion about Figure 2, instead of showing the original images for positive and negative pairs, the author can show the augmented patches. It should be clear that it doesn't make sense to use the 'strong augmentation' since those patches are irrelavant even sampled from the same image.

* In section 2.1, the feature encoder is already defined as f, why not just use f(x) in equation (1) and (2)? Equation (3) is a high level loss function, the left hand side should be something like \theta^{*} where \theta is the learnable parameters in model f.

**Justification Of The Final Rating:**

I appreciate the detailed response from the authors. The hypothesis that this work is based upon: **_there is a distribution difference between natural and medical images in the latent space, so applying the same default strong augmentation settings directly in the pre-training process may not be ideal_** is very straight forward to me. The feature in nature images that determine the class label is usually prominent and takes a good portion of the image. However, in retinal fundus photography, the disease related features can be pretty local, and naturally cropping ratio will significantly effect the training process. Such conclusion can probably be propagated to multiple sclerosis segmentation in brain MRI or any similar topics, but I don't think there is much value in proving this point. Instead, if you can show how to find the optimal cropping ratio for different scenarios (maybe with RL), it will be of more importance.
In general, the paper looks better after revision.

**Justification Of The Preliminary Rating:**

This is an implementation paper, thus there is no technical novelty. The experimental design is problematic such that the results can not prove the assumption. The whole thing needs to be reconsidered and re-worked. I don't see much value to discuss this during the conference.

**Questions To Address In The Rebuttal:**

* What are the SOTA of these downstream tasks? How are they compared with the proposed work?

---

> ### Author Response · Authors · 2025-03-02
>
> Dear reviewer QQYm:
>
> Thank you for your detailed suggestions. I appreciate that you acknowledged this paper contained sufficient visualizations to effectively convey the ideas of this work. We carefully read your suggestions and hoped that our explanations addressed your concerns. If any further questions or clarifications were needed, we were happy to provide additional details.
> 1. It might be because we did not introduce what DINO is in the manuscript. DINO is a contrastive self-supervised learning method with two encoders: a teacher encoder for global crops and a student encoder for local crops (Table 2). It aims to maximize the similarity between the outputs of the student and teacher models. Positive pairs refer to global and local crops from the same image. In the strong augmentation (the default setting in DINO), global crops are patches with a range of (0.4, 1.0), and local crops are patches with a range of (0.05, 0.4). A local crop can be part of a global crop, allowing positive pairs to be constructed and used for pre-training the model. We completely agree that extreme cropping and color jittering can introduce noise into the pre-training process. That’s exactly why we increased the cropping scales and reduced the scale of color jittering in the weak augmentation for positive pair construction.
>
> 2. We defined strong augmentation in the first paragraph of the Experiment: Pre-Training Details section. The strong augmentation followed the default augmentation settings in DINO, which were well-tuned on natural images. You can find the augmentation parameter settings of DINO at: https://github.com/facebookresearch/dino/blob/main/main_dino.py. Our hypothesis is that there is a distribution difference between natural and medical images in the latent space, so applying the same default strong augmentation settings directly in the pre-training process may not be ideal. That’s why we used the default augmentation settings in DINO as strong augmentation. For weak augmentation, we defined it in the second sub-section of the Method section. It was defined as scaling down the strong augmentation (default augmentation settings) to ease the challenge of training the model to converge while also avoiding it being too weak for the model to learn generalizable features. We also provided the specific parameter settings in Table 2. This work aims to validate our hypothesis but does not necessarily require finding the best parameter settings.
>
> 3. Our study focused on hypothesis proposal and validation. It stressed the importance of reducing the augmentation scale when applying contrastive SSL in the medical imaging domain. Comparing our results with the SOTA performance of downstream tasks is outside the scope of our research. But we will explore it in the future and integrate it with other contrastive SSL methods.
>
> 4. The yellow dots represented repeated augmentations of the same image (each augmentation was slightly different due to randomness), and the same logic applied to the blue dots. Before adding any augmentations, the blue and yellow dots were clearly separated. When strong augmentation was applied, these blue and yellow dots had a wider distribution but overlapped heavily with each other in the latent space. When weak augmentation was applied, these blue and yellow dots had only slight overlap. The original Fig. 1c did not show this clearly, so we modified it according to the description above. You can find it in Fig. 1c in our updated manuscript.
>
> 5. That’s a good point! We added augmented patches, as you suggested.
>
> 6. In the first paragraph of this section, we defined the model as ‘f’, which includes two encoders: the student and teacher models, respectively. The encoder is defined as ‘En’.
>
> We hope you can raise your score, so we can make better contributions to this community. If there are any further suggestions regarding this work, we are very willing to explain and provide more details.

---

> > ### Author Response · Authors · 2025-03-11
> >
> > Dear reviewer QQYm:
> >
> > Thank you for engaging in the discussion period, raising the score, and recognizing that our method can likely be extended to multiple sclerosis segmentation in brain MRI or other related topics. Your understanding of our hypothesis is accurate, and it is relative intuitive. Previous studies published in top-tier journals have typically applied the default strong augmentation settings intended for natural images directly to medical imaging applications. For example, Zhou et al. pre-trained several contrastive SSL models on color fundus images, directly applying augmentation settings established for natural images (see the first paragraph of the “Contrastive SSL implementation” section in https://www.nature.com/articles/s41586-023-06555-x ). Similarly, Daniel et al. pre-trained contrastive SSL models on lung CT images without modifying the hyperparameters originally intended for natural images (see the second paragraph of the “Contrastive Learning” section in https://www.nature.com/articles/s41598-023-46433-0.pdf). Previous studies have not investigated the impact of the distribution differences between medical and natural images on the contrastive SSL pre-training process, primarily due to the substantial requirements involved: specifically, a large-scale medical image dataset (e.g., our dataset contains 1.4 million color fundus images) and high-performance graphics hardware (e.g., NVIDIA A100 80GB GPUs in our case). In contrast, our work conducts extensive experiments to quantify how these distribution differences influence pre-training outcomes. Our findings provide evidence that the difference in image distribution affects the pre-training process, thus laying a foundation for future development of medical foundation models.
> >
> > We hope you can raise your score, so we can make better contributions to this community. If there are any further suggestions regarding this work, we are very willing to explain and provide more details.

---

### Official Review · Reviewer_jGQM · 2025-02-22

**Confidence:** 4
**Preliminary Rating:** 4
**Recommendation:** Poster

**Summary:**

The paper explores the hypothesis that weaker augmentation scales enhance contrastive learning for medical imaging by reducing the overlap of augmented images in the latent space, thereby improving model convergence. The authors have done extensive experiments on six retinal datasets, and the study demonstrates that models pre-trained with weaker augmentations achieve significantly better performance in terms of AUPR and sensitivity.

**Strengths:**

1. The authors provide a very interesting and novel hypothesis that weaker augmentation scales can enhance contrastive learning for medical imaging.
2. The experiments shown in this paper are comprehensive, and the evidences to support the hypothesis are robust.
3. This paper is well written with clear structures and explanations.

**Weaknesses:**

1. While the authors have shown weaker augmentation scales can indeed provide better results using DINO, other contrastive learning methods should also be tested.
2. Based on the results from table 3, the performance improvements with weaker augmentations are only slight in some cases, suggesting that the benefits may not be substantial across all datasets and tasks.
3. Authors visually compare the clustering performance with different configurations, but it is better also to include quantitative metrics in comparison.
4. Apart from using in retinal imaging, how can this find be generalized to other medical domains?

**Detailed Comments:**

Please address the issues I mentioned in the weaknesses section.

**Justification Of The Preliminary Rating:**

The authors have provided some interesting findings in the relationship between the augmentation and performance of contrastive learning. While there are certain limitations in evaluation methods and number of contrastive learning methods used for testing, this paper is rather solid in approaching this hypothesis.

**Questions To Address In The Rebuttal:**

Please revise the manuscript based on the weakness section.

**Special Issue:**

No

---

> ### Author Response · Authors · 2025-03-02
>
> Dear reviewer jGQM:
>
> Thank you for your suggestions and for acknowledging that our research provided a very interesting and novel hypothesis with comprehensive experiments to validate it. We carefully read your suggestions and hoped that our explanations addressed your concerns. If any further questions or clarifications were needed, we were happy to provide additional details.
> 1. We agree with this and also highlighted it in the second paragraph of the Discussion section. It was difficult to include more contrastive SSL methods in the experiment due to the page limit. We will incorporate more contrastive SSL methods in future experiments.
>
> 2. In Table 3, 10/18 results showed significant improvement, while 8/18 did not. As mentioned in the second paragraph of the Discussion section, this slight advantage is likely due to the nearly saturated performance after pretraining on large-scale natural images. More importantly, the weak augmentation scale settings are not necessarily optimal, we will explore methods for automatically adjusting the augmentation scale in future work. This work aims to validate our hypothesis but does not necessarily require finding the best parameter settings.
>
> 3. That’s a good suggestion! We used the Silhouette Score (https://ieeexplore.ieee.org/abstract/document/9260048) to evaluate the clustering of feature embeddings in the latent space. The score under strong augmentation is 0.117, under weak augmentation is 0.201, and under weak+med augmentation is 0.130. We added it to the first paragraph of the Experiment Results section.
>
> 4. Indeed, we used only one modality. However, the hypothesis we proposed is domain-agnostic and not limited to a single modality. We believe it can be applied to all medical imaging modalities, and we will include more modalities in future studies.
>
> We hope you can raise your score, so we can make better contributions to this community. If there are any further suggestions regarding this work, we are very willing to explain and provide more details.

---

> > ### Comment · Reviewer_jGQM · 2025-03-14
> > **Reply to the authors**
> >
> > Thank you for addressing my concerns and answering the questions. I do not have any other questions.

---

> > > ### Author Response · Authors · 2025-03-14
> > >
> > > We are really glad to know that we have addressed your questions. Thank you again for your insightful suggestions. If possible, we would greatly appreciate it if you could consider raising the score. Thanks!

---

> ### Comment · Area_Chair_MVSd · 2025-03-17
> **Final rating is missing**
>
> Dear jGQM,
>
> Would you be so kind to provide your final rating? Thanks!
>
> - AC

---

### Author Response · Authors · 2025-03-07

Dear reviewers and AC:

Thank you for your effort in reviewing our manuscript and providing constructive suggestions. We appreciate that some of you recognized our work for its novel hypothesis and comprehensive experiments, as well as acknowledged the efficacy of our method. We have carefully reviewed all your suggestions and provided point-to-point responses below.
1. For Reviewer jGQM: We added a metric to quantify the model’s clustering quality. We also explained why weak augmentation did not show a strong advantage in some cases and clarified that our work is domain-agnostic, making it applicable to other modalities. We will explore more contrastive SSL models in future work.

2. For Reviewer QQYm: We clarified the definitions and settings of weak and strong augmentations and highlighted that our work proposed an important hypothesis and validated it with extensive experiments. Our goal is to optimise the contrastive learning training, particularly considering the inherent differences between natural images and medical images, as an efficient way to improve the clinically-relevant applications, rather than outperforming state-of-the-art performance.

3. For Reviewer BvG5: We discussed why medical augmentations might not be beneficial in our experiments. We also clarified why the benefit of weak augmentation is not as strong in the external experiment. We have also added a clustering metric and relevant references as suggested.

We are keen to engage with all the reviewers for further clarification. Thank you.

---

> ### Author Response · Authors · 2025-03-14
> **Summary on rebuttal and discussion**
>
> Dear reviewers and ACs:
>
> Thank you for actively engaging in the discussion. We are pleased that reviewers jGQM and BvG5 felt all concerns were addressed (**weak accept**). Additionally, reviewer QQYm acknowledged improvements in the paper and its potential generalisability ("such a conclusion can probably be propagated to multiple sclerosis segmentation in brain MRI or similar topics"), and accordingly raised the score (**weak reject**).
>
> We would like to further emphasise the main contribution of our **hypothesis-validation study**. Previous works in top-tier journals have typically applied default strong augmentation settings (originally tuned on natural images) directly to medical images in large-scale pre-training, lacking investigation of how distribution differences between medical and natural images impact pre-training outcome. **In this study**, we systematically quantified how these distribution differences influenced pre-training outcomes, using real-world large-scale medical images (i.e. 1.4 million colour fundus images) and high-performance computing infrastructure (i.e. NVIDIA A100 80GB GPUs). The results demonstrated that a weaker augmentation is more suitable to medical images, providing a solid foundation for optimising the hyperparameters for future medical foundation model research.
>
> Bests,
> All authors

---

### Author Rebuttal · Authors · 2025-03-07

**Rebuttal:**

Dear reviewers and AC:

Here is the revised manuscript. We have highlighted the changes in red. If any further questions or clarifications were needed, we were happy to provide additional details.

**Supporting Material:**

/attachment/3e951b49826a454977ce30cf133475f5e0bf177a.pdf

---

### Comment · Area_Chair_MVSd · 2025-03-08
**Time for discussion and review of the rebuttal**

Dear reviewers

It is now time to consider the responses from the authors. If you are or are not satisfied with author's reply please still post to openreview your feedback to the rebuttal and update your scores. Especially, please update the scores if you feel that the authors have addressed your concerns.

Please note that you can and **are encouraged** to discuss the scores of other reviewers if you disagree with them to make the best

As AC, my responsibility is to post meta-reviews by March 21st, and I would thus like to kindly ask you to consider the authors' rebuttal as soon as possible.

// Your Area Chair

---

### Meta-Review · Area_Chair_MVSd · 2025-03-24

**Recommendation:** Accept (Poster)
**Confidence:** 4

**Metareview:**

This paper looks at how augmentation strengths affects self-supervised learning quality in retinal images. The reviewers are generally leaning toward accepting, except for one reviewer.

I have read the paper and concluded that this work has merit, albeit authors making too general statements (first 50% of the abstract, e.g.  are not discussing RI at all). Perhaps, making sure that the readers do not feel this way during camera-ready would be a good idea.

I note that the claim of reviewer of **QQYm** has merit: the natural images and the retinal images are different (see e.g. works by Konz et al on intrinsic manifolds). Having said that, this paper gives an insight, and the study seems to be well-done and has rigorous reporting.